# Radiation Hardness Study of Silicon Carbide Sensors under High-Temperature Proton Beam Irradiations

**DOI:** 10.3390/mi14010166

**Published:** 2023-01-09

**Authors:** Elisabetta Medina, Enrico Sangregorio, Andreo Crnjac, Francesco Romano, Giuliana Milluzzo, Anna Vignati, Milko Jakšic, Lucia Calcagno, Massimo Camarda

**Affiliations:** 1Physics Department, Università degli Studi di Torino, Via Pietro Giuria 1, 10125 Turin, Italy; 2INFN–National Institute for Nuclear Physics, Turin Division, Via Pietro Giuria 1, 10125 Turin, Italy; 3STLab srl, Via Anapo 53, 95126 Catania, Italy; 4Dipartimento di Fisica e Astronomia Ettore Majorana, Università degli Studi di Catania, Via S. Sofia 64, 95123 Catania, Italy; 5Istituto per la Microelettronica e Microsistemi IMM–CNR, Sezione di Catania, Strada VIII Zona Industriale 5, 95121 Catania, Italy; 6Division of Experimental Physics, Ruđer Bošković Institute, 10000 Zagreb, Croatia; 7INFN–National Institute for Nuclear Physics, Catania Division, Via S. Sofia 64, 95123 Catania, Italy; 8SenSiC GmbH, DeliveryLAB, 5234 Villigen, Switzerland

**Keywords:** silicon carbide, radiation hardness, proton irradiation, high-temperature irradiation

## Abstract

Silicon carbide (SiC), thanks to its material properties similar to diamond and its industrial maturity close to silicon, represents an ideal candidate for several harsh-environment sensing applications, where sensors must withstand high particle irradiation and/or high operational temperatures. In this study, to explore the radiation tolerance of SiC sensors to multiple damaging processes, both at room and high temperature, we used the Ion Microprobe Chamber installed at the Ruđer Bošković Institute (Zagreb, Croatia), which made it possible to expose small areas within the same device to different ion beams, thus evaluating and comparing effects within a single device. The sensors tested, developed jointly by STLab and SenSiC, are PIN diodes with ultrathin free-standing membranes, realized by means of a recently developed doping-selective electrochemical etching. In this work, we report on the changes of the charge transport properties, specifically in terms of the charge collection efficiency (CCE), with respect to multiple localized proton irradiations, performed at both room temperature (RT) and 500 °C.

## 1. Introduction

One of the main characteristics required for sensors used in diagnostic applications is the capability of withstanding harsh environment (HE) operations. Examples of HE include (i) X-ray sensors in the extreme-intensity beams of synchrotrons and free-electron lasers with beam powers exceeding 100 kW/cm${}^{2}$, (ii) electron sensors in sterilization processes and novel radiotherapies, and (iii) neutron sensors for safety assessments and process monitoring in nuclear facilities. The high total radiation doses, as well as the instantaneous ones, to which these sensors are exposed require stable and reliable responses over long periods of time, possibly even under high-temperature operation conditions. Nowadays, solid-state technology represents a solution for several sensing applications, thanks to, among other factors: high signal-to-noise ratio, small size, lateral resolutions, and fast response time. However, radiation hardness is a critical characteristic for solid-state sensors. Silicon, due to its low bandgap and low kick-off energy, cannot generally be used in any HE applications. Diamond, on the other hand, due to its very large bandgap, maximum theoretical operating temperature, and very high kick-off energy, is the most studied semiconductor, in particular with studies focused on electronic response at high device temperature. However, the very high cost, limited sample size (less than 1 cm${}^{2}$ for CVD single crystal diamond), high level of impurities, and physical limitations on doping control have prompted the scientific community to find alternative solutions. Silicon carbide (SiC), a semiconductor composed of 50% silicon and 50% carbon atoms, represents the most obvious candidate, compromising between the low cost and industrial maturity of silicon, on one hand, and the radiation-hardness capabilities close to diamond, on the other. In the last 5–10 years, SiC has been the protagonist of industrial advances, and the reason for this maturity is its wide use for power electronics applications, particularly in high-end electric vehicles. Despite numerous studies in terms of high-power applications, the characteristics of SiC sensors, especially in high-temperature operations, require specific experiments and investigations, including the one described in the following paragraphs.

Previous research has demonstrated, among other things, the detection of alpha particles with SiC-based detectors at high temperatures (up to 500 °C), strengthening the potential of these devices for multiple applications in harsh environments [1]. Furthermore, it is now well known that ion implantations, the only method to achieve selective doping of SiC regions in power devices, induce an accumulation of point defects which, for high injection doses, can even lead to amorphization. However, high-temperature (500 °C and above) implantations hinder this effect, thanks to a dynamic annealing process occurring under these conditions. More details regarding ions implantation at high temperature in silicon carbide can be found in [2,3,4,5,6,7]. Consequently, aware of the good functionality of SiC above 500 °C and of the high temperature condition needed during ion implantations, it may be a valid solution, for some applications in extremely harsh environments such as within fusion reactors core vessels, to employ SiC sensors at high temperatures to make them more radiation tolerant. An experimental result consistent with this approach is explained in [8], where a high-power 4H-SiC Schottky diode was irradiated with electrons and showed a drop of the carrier removal rate by about six orders of magnitude in the case of 500 °C irradiations. The aim of the experiment described in this paper is to study the effect of high-temperature irradiation on an SiC device, comparing the results with those obtained in [8] in terms of sensor functionality (i.e., local CCE instead of reverse leakage measurement), and with a different beam condition: a focused proton beam instead of a large-area electron one. In more detail, for this work, proton beams with MeV energies were used either to inject charge carriers (probing ion beam) or induce radiation damage (damaging ion beam) in SiC sensors. Damage was induced at both room and high temperatures (>400 °C). Probing was performed at RT. This activity, focusing on proton-induced charge transport properties, is part of a more general effort to establish the functionality of SiC with different ionizing radiation beams and environments, in order to validate their possible use in a wide range of applications. The properties of sensors depending on temperature and device geometries (i.e., “bulk type” or “independent membrane type”, see below), which are used to assess the safety ranges for the operational functionality of SiC-based sensors, have been studied. The capability of the selected facility to expose small areas (below 100 × 100 $\mu m^{2}$) with the beam, thus evaluating local effects within a single device and reducing uncertainties generated by the device-to-device variabilities, has been exploited.

## 2. Materials and Methods

The Ruđer Bošković Institute (RBI [9]) accelerator facility consists of two accelerators, 6.0 and 1.0 MV electrostatic tandem accelerators (6.0 MV EN Tandem Van de Graaff and 1.0 MV HVE Tandetron), as well as nine beam lines, represented schematically in Figure 1. In this experiment, the SiC sensors were mounted in an ion microprobe vacuum chamber, connected to one of the beamlines, in which a system of quadrupole lenses (depending on the application, it could be doublet, triplet, or quintuplet) is able to focus the accelerated ion beam to the micrometer size. The Beam-Induced Charge Technique (IBIC) was exploited: fast ions crossing the sensor interact with the electrons of atoms in the material, and numerous ionizations along their trajectories are created. The interaction of the ions with the semiconductor device will eventually generate electron–hole (e–h) pairs, which can drift due to the built-in electric field (e.g., pn junction) or to an externally applied electric field, generating a measurable current signal at the electrodes. During the irradiation, the ion beam, thanks to the micrometer size, can be scanned over the desired sample regions, so that the collected signal can be correlated to the beam position, enabling 2D mapping of charge transport properties (IBIC maps).

In the experiment described in this work, an SiC membrane sensor was used, produced by the SenSiC company [10]. SiC membranes have recently demonstrated promising hard X-ray beam position monitoring capabilities [11], as well as promising ultra-high-dose-rate electron beam dosimetry monitoring capabilities, in the so-called Flash Radiotherapy application [12,13]. These devices are semiconductor PiN junctions: they are composed of a thin, 0.3 μm p${}^{+}$ highly doped ($10^{18}$ cm${}^{-3}$) layer and a 20 μm n${}^{-}$ low-doped ($10^{14}$ cm${}^{-3}$) layer on top of a ∼370 μm thick n${}^{+}$ ($10^{18}$ cm${}^{-3}$) substrate. The n${}^{+}$ substrate of the sample used in the experiment has been partially removed by electrochemical etching (with the expertise of STLab [14] and SenSiC), creating a thinned-down circular area in a selected region of the sensor (i.e., a free-standing membrane) [15]. The total sensor area (5×5 mm${}^{2}$) is divided into four pads (2.5×2.5 mm${}^{2}$ each), and the central area at the four pads (circular region of ∼2 mm diameter) is thinned using an electrochemical doping-selective etching. On the rest of the sensor, the 370 μm bulk is still present under the active layer. The cross-section of the sensor used is shown in Figure 2. Only one of the four pads was connected to the electronics and studied in the Ion Microprobe Chamber.

Since the device was tested under simultaneous exposure to heat and ion beams, good thermal resilience of the sensor, as well of the signal-processing components, in the vacuum chamber had to be ensured. High-purity silver paste was used to mount the SiC sensor on the ceramic PCB, which was in direct contact with a heating element mounted in the vacuum chamber. Details of this setup can be found in [16]. Reverse bias, up to −100 V, was applied through the front electrode, while the back electrode was grounded, thus achieving “reverse diode operation”, typical for achieving a high signal-to-noise ratio in sensing applications. To make the temperature rise possible, the sensor was heated by a resistive heater, contacting the ceramic plate, and the temperature was checked by means of a type K thermocouple. The electronic readout chain processing the sensor’s output signal comprises a charge-sensitive preamplifier (ORTEC 142A), a spectroscopy amplifier (ORTEC 570), an analog-to-digital converter (Canberra8075) module, and the in-house-developed SPECTOR software [17]. Finally, a source-measurement unit (SMU, Keithley 6485 pico-ammeter) with a current range of 2 nA–2 mA and 10 fA of resolution was used to characterize the current-voltage curve of the device in the dark (i.e., without beam).

The irradiation and subsequent charge collection study was performed in the Ion Microprobe Chamber already described, using a proton beam focused down to the smallest size of ∼1 μm radius. The micrometer beam can be scanned over the sensor surface in specific selected areas, with the possibility of choosing both the scanning speed and size of the scan. During the experiment, two different linear accelerators were used, depending on the specific irradiation purpose. A low-energy proton beam (1 MeV) was used for the IBIC studies, while a 3.5 MeV proton beam (delivered by the larger of the two accelerators) was used to generate damaged regions. The lower beam energy was used for charge collection efficiency (CCE) measurements. In this case, simulations with SRIM software [18] demonstrated that the energy is deposited completely within the 20 μm active thickness, with a Bragg peak located around half of the active area (∼ at 10 μm depth). On the other hand, the 3.5 MeV proton beam was used to locally induce radiation damage in selected areas of the sensor. In this case, the beam passes completely through the sensor, creating almost homogenous defects along the trajectory with an average of ∼2 $\times 10^{-5}$ vacancies produced per ion per micrometer of penetration. It was verified by simulations that only ∼10% of the energy is released in 20 μm and that the Bragg peak is located within the substrate. Both simulation results are shown in Figure 3.

To calibrate the charge collected by the silicon carbide during IBIC tests, a reference silicon STIM (scanning transmission ion microscopy) detector fixed inside the chamber under the same beam conditions is used, assuming a total collection (100%) of the beam signal. The channel-to-energy calibration is implemented as shown in Formula (Equation 1), where $W_{\mathrm{SiC}}$ and $W_{\mathrm{Si}}$ are the average energies for e–h pair creation in SiC and Si (7.28 eV and 3.62, respectively), $CHN_{\mathrm{Si}}$ is the channel corresponding to the energy peak measured by the fixed silicon sensor, and β is a correction factor containing any differences in the gain set in the electronics: (1)$$Energy=CHN_{\mathrm{SiC}}\cdot \left(\frac{W_{\mathrm{SiC}}}{CHN_{\mathrm{Si}}\cdot W_{\mathrm{Si}}}\cdot \beta \right)$$

Depending on the energy of the proton beam, a different number of e–h pairs is created, which move through the sensor, generating a measurable signal. In the presence of defects along the path, the charge carriers can be trapped in the crystal lattice and consequently reduce the signal. There are mainly two mechanisms of radiation-induced degradation in a semiconductor device, as explained in detail in [8]: the creation of deep acceptor levels (to which electrons pass from shallow donor levels) and the interaction of radiation defect (vacancy) with a shallow impurity atom to give an electrically neutral (or acceptor) center. Models have showed that the first mechanisms leads to a linear fall of carrier concentration with increasing irradiation dose, while in the second case, the decrease is exponential. Experimental data from different researchers [19,20] have demonstrated that the first mechanism is dominant in SiC devices. Therefore, with the increase of the irradiation dose, a linear degradation of SiC performance is expected. However, in the case of high-temperature irradiation, as has already been mentioned, there is the possibility of increased radiation tolerance due to less lattice damage under such conditions. The next section describes the results obtained from charge collection efficiency measurements, which are consistent with this scenario.

## 3. Results and Discussion

For the current-voltage characterization of the device, the previously described pico-ammeter was used. By measurements at room temperature in dark conditions, it was verified that up to −60 V of bias voltage, the leakage current does not exceed 1 nA, while at −80 V, this reaches 8.5 nA, as is shown in Figure 4. We decided not to overcome this latter voltage value (corresponding to an electric field of 4 V/μm in the active thickness) because thereafter the leakage current would rise considerably. We performed the measurement at the very beginning and after all tests to verify the proper functionality of the device and to check that the irradiations had not affected the electronic performances. This result indicates that in the chosen range of bias voltages, the device remains in a safe condition and the leakage current is negligible compared to the signal.

Before starting the sensor damage test and the subsequent charge collection study, we studied the spatial uniformity of the charge collection in the sensor. By scanning the micrometric beam over a desired area, a homogeneous charge collection efficiency was recorded throughout the active area of the sensor. Moreover, the presence of the membrane and the boundary between the membrane and the rest of the sensor with the transmission beam were observed and identified. Protons of 3.5 MeV have a penetration depth of more than 20 μm (which is the thickness of the active region throughout the device, both in and outside the membrane portion). Charge carriers generated in regions deeper than 20 μm can still reach the active layer through diffusion processes and contribute to the induced charge signal. The overall contribution of the diffusion component is expected to be small compared to primary drift, therefore the relative difference in CCE between two regions is also small, less than 10%. Thus, the acquired IBIC maps with 3.5 MeV beam were subsequently used to define regions of interest inside or outside the sensor membrane area for probing or inducing radiation damage. For this, small regions of the same sensor were selected. In fact, the facility makes it possible to study the effect of different irradiations on the same sample while keeping the overall properties and removing the uncertainty that would exist when using different sensors for different tests. Several square areas on the sensor surface were selected: four to be irradiated at room temperature and four to be irradiated at high temperature, using different beam fluences. Table 1 describes the fluences used to irradiate the sensor, the respective vacancies’ densities, and the respective doses. The dose was obtained by multiplying the stopping power of a 3.5 MeV proton beam in water (107.4 MeV cm${}^{2}$g${}^{-1}$) by the irradiation fluence with which the sensor was damaged. During the irradiation, the beam was periodically intercepted (chopped) with a gold-plated aluminum sheet. The backscattered spectra collected from the chopper was then used to estimate the total deposited fluence during the irradiation. The chopper calibration setup was described in [16], and the estimated statistical error of the number of ions is better than 5%. The beam was focused to a micrometer spot size and scanned over the desired rectangular sensor region. Therefore, both the number of ions and irradiation area size were well defined and used to estimate the irradiation fluence.

Initially, we irradiated the four zones by subjecting the sensor to high temperature (500 °C), and then the operation was repeated at room temperature, selecting different areas from the previous ones. Figure 5 shows an IBIC map (at 10 V) acquired after all induced damage. The irradiated regions of the sensor are evidently identifiable as they have lower CCE compared to unirradiated sensor. The drop in CCE is proportional to the ion fluence deposited in the different regions, so different colors in Figure 5 correspond to different ion fluences. Under these conditions, it is not possible to distinguish the membrane from the bulk because the Bragg peak for 1 MeV protons is around 10 μm under the surface, well within the 20 μm active layer, so the contribution from carriers generated in the substrate (i.e., further than 20 μm from the surface) is negligible. The regions identified in the figure with A and B correspond to regions on the bulk of the sensor (at room and high temperature). Zones C and D are located on the membrane.

The charge collection efficiency, obtained with the IBIC technique at room temperature with the lower-energy beam (1 MeV), in the differently irradiated areas has been studied to observe the effect of the different damaging conditions. Furthermore, the signal was studied at different biases, in a range between 0 V and 60 V, by steps of about 5 V each. The overall results are shown on the right in Figure 5, where data obtained from a non-irradiated area (labeled as pristine) are also shown. The trapping time is inversely proportional to the charge carrier drift velocity [21], the relative difference in collection efficiency due to different amounts of radiation damage become less significant at higher drift velocities, and CCE saturates to some constant values. A total charge collection (100%) is never achieved, and this is due to the basic assumption we made for the reference silicon STIM detector used for calibration: it may not collect exactly 100% of the charge, and this would lead to an increase in the CCE evaluated here. As expected, the higher the irradiation fluence, the worse the charge collection efficiency. The results demonstrate also, for the first time to our knowledge, the effects of the dynamic annealing on the CCE, proving higher collection efficiencies in the areas damaged at high temperature (empty markers) compared to areas damaged at room temperature (filled markers) for all fluences tested, indicating that the radiation damage induced at elevated temperatures has suppressed the influence on the charge collection efficiency of the sensor. This result supports our previous assumption: dynamic annealing effects at temperatures around 500 °C in silicon-carbide crystal lattice leads to lower overall accumulation of radiation damage due to enhanced mobility of radiation-induced point defects [22,23]. Subsequently, this results in a lower decrease in the charge collection of the sensor. The difference in CCE between those areas varies between about 20% (for low voltages) and 5% for higher voltages applied. Charge collection efficiency always exceeds 80% above 30 V (except for the higher fluence damaging). This indicates a good recovery despite the damages, except for the two most intensely damaged zones (in red). In the case of the lowest fluence used ($5\times 10^{12}$ protons/cm${}^{2}$), the curve gets up to 90% of the CCE, reaching the same values as the undamaged area.

One of the potential features of this sensor is the presence of the thin membrane (20 μm in this case). The next goals in our work include characterizing the membrane, comparing its performance with the bulk, and investigating the effects of this on the internal electric field and thus on its charge transport properties. In these tests, all damages were done at room temperature. A first step in this future work was done at the RBI in conjunction with the tests just described. To this end, seven more zones were selected in another region on the sample, four within the membrane (F zones) and three on the bulk (E zones), and irradiated at room temperature using the beam parameters described in Table 2. The total IBIC map measured after the irradiation is shown in the left panel of Figure 6.

Observing the curves on the right panel of Figure 6, where the CCE as a function of the applied bias voltage to the sensor for the different irradiation conditions (also two non-irradiated regions labelled as pristine) is reported, it is clear that the areas selected over the membrane (cross markers) measure a higher charge (higher CCE) with respect to the areas irradiated over the region with the bulk (triangle markers) at the same irradiation fluence. This result may be partially caused by backscattering events coming from the bulk, which may create more damage on the active thickness of these regions. However, the main reason is more likely to be related to the electric fields established with/without the bulk, which may modify the carrier transport properties. This phenomenon of radiation hardness being influenced by the sensor thickness needs to be confirmed with further experimental efforts; however, it opens a possibility of new applications of SiC membrane sensors operated in harsh radiation conditions.

## 4. Conclusions

In summary, an SiC membrane sensor with a thickness of 20 μm on the membrane (and 370 μm on the bulk) was subjected to several irradiation tests at the Ruđer Bošković Institute, specifically within the Ion Microprobe Chamber facility. The ability to deliver a micrometer beam was exploited to study the transport properties of the sensor locally, subjecting the sample to multiple irradiation doses at two temperatures (RT and up to 500 °C). The uniformity of the sensor surface fabricated by SenSiC was verified for the first time by observation of homogeneous collected charge over the studied areas. After different irradiations, higher charge collection efficiency was observed for areas damaged at high temperatures than those irradiated at room temperature, suggesting that SiC radiation hardness might be further enhanced by operating the sensors in high-temperature operating conditions. Furthermore, preliminary results suggest that sensors on free-standing membranes could benefit from higher radiation hardness as compared to standard ‘bulk’ ones. Additional beamtimes in the facility are planned, both to further investigate membrane properties and to test other comparable sensors with ultra-thin membranes (2 μm and 200 nm are already available for testing). Simulations will also be needed to study the physical phenomena that regulate the behavior of charges along the membrane subjected to ultra-high-dose deposition beams. 

## Figures and Tables

**Figure 1 micromachines-14-00166-f001:**
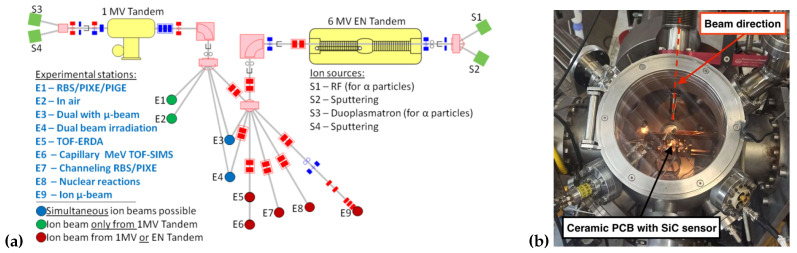
(**a**) Beamlines overview at the RBI accelerator facility [9]. THE 6.0 MV EN Tandem Van de Graaff and 1.0 MV HVE Tandetron are represented in yellow. The E9 experimental station corresponds to the Ion Microprobe. (**b**) The Ion Microprobe Chamber is shown, in which the beam is focused down to micrometer spot size, and in which the sample to be tested is placed (the photo shows the SiC membrane on the support structure).

**Figure 2 micromachines-14-00166-f002:**
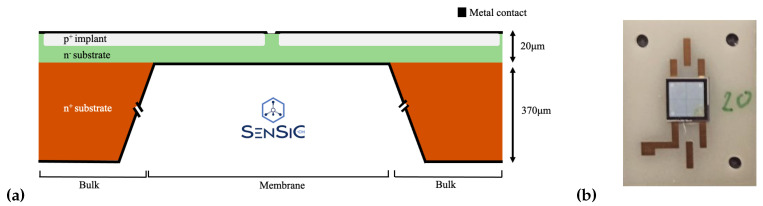
(**a**) The structure of the SiC membrane sensor is shown: a thin p${}^{+}$ layer, an n${}^{-}$ layer, and a thick n${}^{+}$ substrate. The central area at the four pads (circular region of ∼2 mm diameter) is thinned using an electrochemical doping-selective etching. (**b**) Photo of the sensor mounted on a ceramic plate with gold electrodes, subsequently mounted inside the chamber.

**Figure 3 micromachines-14-00166-f003:**
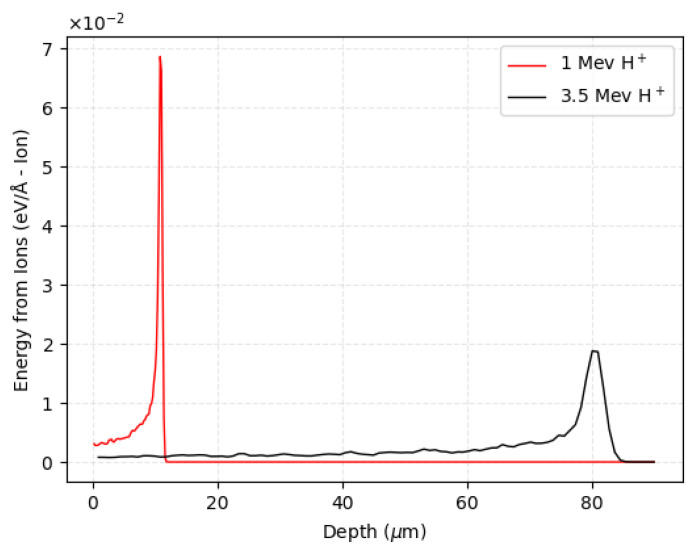
Energy loss as a function of depth in silicon carbide for a 1 MeV (in red) and 3.5 MeV (in black) protons beam produced by the SRIM simulation software.

**Figure 4 micromachines-14-00166-f004:**
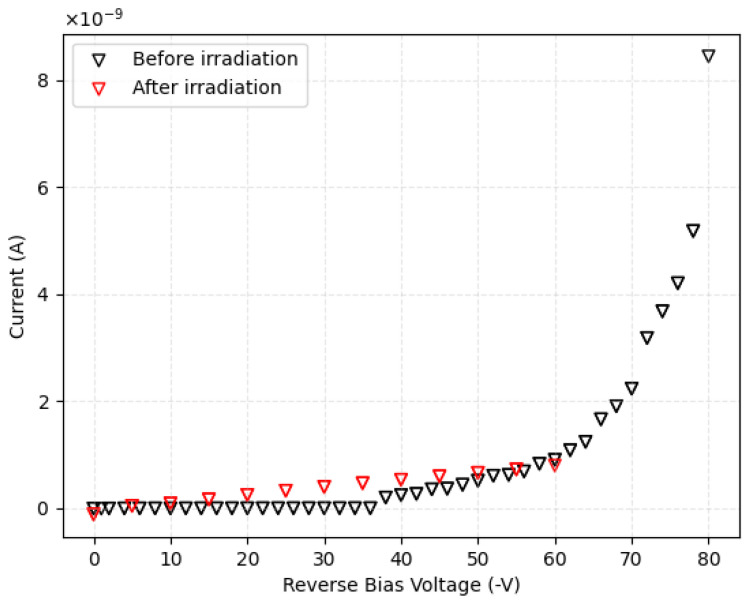
Dark I–V characterization. The curve before any test (irradiation and IBIC) is represented in black, while the curve after all irradiations (both at room temperature and at high temperature) is represented in red.

**Figure 5 micromachines-14-00166-f005:**
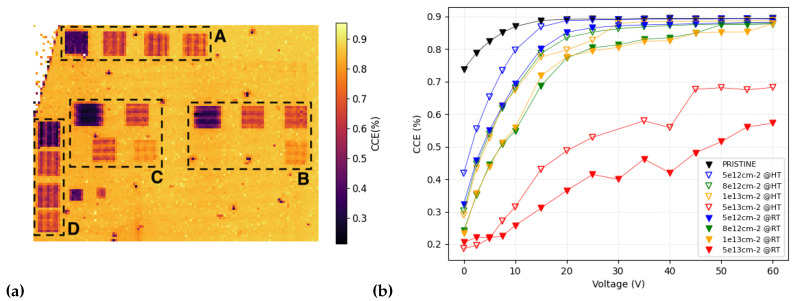
(**a**) IBIC map acquired at 10 V on the sensor region containing the damaged areas. The framed areas in the A region were irradiated at room temperature, and the B areas at 500 °C, both on the bulk. The C and D areas are located on the membrane irradiated at room temperature and 500 °C, respectively. The color-bar on the right shows the CCE values. (**b**) CCE as a function of the applied bias voltage. Different colors indicate different irradiance fluences, full markers indicate measurements performed in A region (room temperature), while empty ones indicate B region (500 °C).

**Figure 6 micromachines-14-00166-f006:**
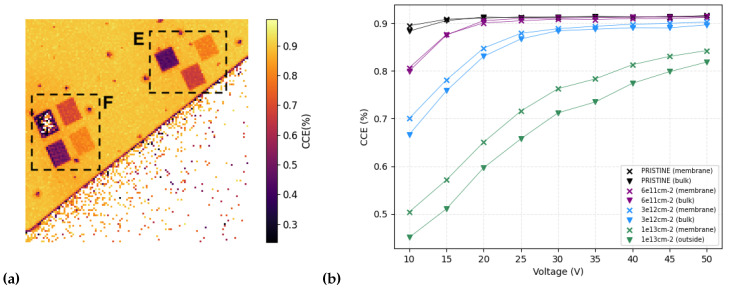
(**a**) IBIC map acquired at 10 V on the sensor region containing the damaged areas. The framed areas in the E region were in the bulk, and the F areas were on the membrane. The colorbar on the right shows the CCE values. (**b**) Charge collection efficiencies as a function of the applied bias voltage. Different colors indicate different irradiance fluences, triangle markers indicate E regions, and crosses indicate F regions.

**Table 1 micromachines-14-00166-t001:** Fluences used to irradiate the sensor at both room and high temperatures, with a 3.5 MeV proton beam (transmission beam). Four fluences were chosen at both room temperature and high temperature. Respective vacancies’ densities and the doses are also reported.

	First Fluence	Second Fluence	Third Fluence	Fourth Fluence
Fluence (cm${}^{-2}$)	$5\times 10^{12}$	$8\times 10^{12}$	$1\times 10^{13}$	$5\times 10^{13}$
Vacancies density (cm${}^{-3}$)	$9.94\times 10^{11}$	$1.59\times 10^{12}$	$1.99\times 10^{12}$	$9.94\times 10^{12}$
Dose (Gy)	$8.6\times 10^{4}$	$1.38\times 10^{5}$	$1.72\times 10^{5}$	$8.6\times 10^{5}$

**Table 2 micromachines-14-00166-t002:** Fluences used to irradiate the sensor inside the membrane and on the bulk, with a 3.5 MeV proton beam (transmission beam). Four fluences were chosen, and the highest one was used just on the membrane. Respective vacancies’ densities and the doses are also reported.

	First Fluence	Second Fluence	Third Fluence	Fourth Fluence
Fluence (cm${}^{-2}$)	$6\times 10^{11}$	$3\times 10^{12}$	$5\times 10^{12}$	$1\times 10^{13}$
Vacancies density (cm${}^{-3}$)	$1.19\times 10^{11}$	$5.96\times 10^{11}$	$9.94\times 10^{11}$	$1.99\times 10^{12}$
Dose (Gy)	$1.03\times 10^{4}$	$5.16\times 10^{4}$	$8.6\times 10^{4}$	$1.72\times 10^{5}$

## Data Availability

Not applicable.

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
