# Peer review of "Radiation Hardness Study of Silicon Carbide Sensors under High-Temperature Proton Beam Irradiations"

_micromachines, 2023, doi:10.3390/mi14010166_

Round 1

Reviewer 1 Report

The paper is of high value and can be published after minor improvements. Here are my comments:

1) Explanation of figure 4 is slightly weak and it is difficult to understand the reason why this is important to be included in the manuscript.

2) In tables 1 and 2 vacancies density is converted to Dose (Gy). Please elaborated the conversion procedure.

Author Response

Dear Reviewer,

Thanks a lot for your revision and for your useful suggestions. We revised the paper following both all your and other reviewer suggestions and we are confident our new version is right for publication. Please see the attachment to view our responses to your comments.

Kind Regards

The Authors

Reviewer 2 Report

The article studies the radiation tolerance of SiC in membrane/bulk forms at room and 500℃ temperatures. This study shows experimental proof of better charge collection efficiency in 3.5 MeV irradiated bulk SiC sensor at 500℃ compared to room temperature using a 1 MeV IBIC. However, this article needs improvement in the following areas,

1.     In abstract line 4, radiation tolerance may be a better term instead of radiation resistance. (Also repeated in other sections in the manuscript)

2.     In fig. 1a maybe an arrow and a label are needed. Hard to fig. out the SiC sensor in the chamber.

3.     Is it possible to share more details about electrochemical etching or any references? It can be useful for readers who wish to replicate.

4.     In fig. 4, do the authors mean that the I-V curves are the same at both RT and HT? if so, please clearly indicate it in the description.

5.     The reason given for no differentiation between bulk vs membrane in IBIC as ‘the energy deposited over the bulk is the same as for the membrane.’ Or is it because the Bragg peak for 1 MeV is near the P-N junction in both membrane and bulk cases?

6.     In line 202, the non-irradiated (pristine) area charge collection efficiency is at RT or HT? if both conditions show similar results please indicate.

7.     In fig 5a, there are 4 different colors in the IBIC map is it due to 4 different fluence? If so, please mention, if not, what is that?

8.     Line 210-215: ‘lower creation of radiation damage’ does not sound like the best description. It could be damage creation followed by damage repair but not plain lower damage creation.

9.     Is there any other reference in any other material systems on dynamic annealing at high temperatuer irradiation conditions? If so please include it.

10.  Line 216: What’s the physics behind ‘The difference in CCE between those areas varies between about 20% (for low voltages) and 5% for higher voltages applied. Charge collection efficiency always exceeds 80% above 30 V (except for the higher fluence damaging)’ please explain.

11.  Line 220: what is the 1st irradiation the authors are talking about? This sentence is unclear.

12.  Line 223: the membrane vs bulk characterization at what temperature? Is it RT or HT? no info. on that. (also, in fig. 6b)

13.  Line 226: Is E-zone the BULK? consistent use of outside/bulk should be followed or please clear up what bulk/outside mean.

14.  Any reason why the surface has higher CCE than the bulk? Probably because the Bragg peak for 3.5 MeV in bulk where most damage/energy loss happens creates more defects in bulk than membrane?

15.  Why IBIC maps at 10v in fig. 6a and at 7.5v in fig. 5a? why can't the authors show at same bias voltage in both conditions?

16.  In fig. 5a, 4 different conditions, bulk vs membrane at RT and HT are shown. But the temperature effects description is only for bulk. Why not describe the temperature effects on the membrane? If it is not any different, please show data and describe so!

Author Response

(The authors gave the same response as above.)

Round 2

Reviewer 2 Report

Thanks for addressing most of my comments.

For comment #16, I think there might be some benefit to adding a statement that 'the trends between LT and HT CCE for membrane vs bulk is similar' or something of that sort.